# The Iron Chelator Desferrioxamine Increases the Efficacy of Bedaquiline in Primary Human Macrophages Infected with BCG

**DOI:** 10.3390/ijms22062938

**Published:** 2021-03-13

**Authors:** Christina Cahill, Fiona O’Connell, Karl M. Gogan, Donal J. Cox, Sharee A. Basdeo, Jacintha O’Sullivan, Stephen V. Gordon, Joseph Keane, James J. Phelan

**Affiliations:** 1TB Immunology Group, Department of Clinical Medicine, Trinity Translational Medicine Institute, Trinity College Dublin, St James’s Hospital, 8 Dublin, Ireland; cahillch@tcd.ie (C.C.); gogank@tcd.ie (K.M.G.); DOCOX@tcd.ie (D.J.C.); basdeos@tcd.ie (S.A.B.); josephmk@tcd.ie (J.K.); 2Department of Surgery, Trinity Translational Medicine Institute, Trinity College Dublin, St James’s Hospital, 8 Dublin, Ireland; oconnefi@tcd.ie (F.O.); osullij4@tcd.ie (J.O.); 3UCD Conway Institute of Biomolecular and Biomedical Research, University College Dublin, 4 Dublin, Ireland; stephen.gordon@ucd.ie

**Keywords:** antimicrobials, tuberculosis, BCG, iron metabolism, iron chelation, host-directed therapy, drug-resistant tuberculosis, interferon-γ

## Abstract

For over 50 years, patients with drug-sensitive and drug-resistant tuberculosis have undergone long, arduous, and complex treatment processes with several antimicrobials. With the prevalence of drug-resistant strains on the rise and new therapies for tuberculosis urgently required, we assessed whether manipulating iron levels in macrophages infected with mycobacteria offered some insight into improving current antimicrobials that are used to treat drug-resistant tuberculosis. We investigated if the iron chelator, desferrioxamine, can support the function of human macrophages treated with an array of second-line antimicrobials, including moxifloxacin, bedaquiline, amikacin, clofazimine, linezolid and cycloserine. Primary human monocyte-derived macrophages were infected with Bacillus Calmette-Guérin (BCG), which is pyrazinamide-resistant, and concomitantly treated for 5 days with desferrioxamine in combination with each one of the second-line tuberculosis antimicrobials. Our data indicate that desferrioxamine used as an adjunctive treatment to bedaquiline significantly reduced the bacterial load in human macrophages infected with BCG. Our findings also reveal a link between enhanced bactericidal activity and increases in specific cytokines, as the addition of desferrioxamine increased levels of IFN-γ, IL-6, and IL-1β in BCG-infected human monocyte-derived macrophages (hMDMs) treated with bedaquiline. These results provide insight, and an in vitro proof-of-concept, that iron chelators may prove an effective adjunctive therapy in combination with current tuberculosis antimicrobials.

## 1. Introduction

Antimicrobial therapy for patients with drug-sensitive and drug-resistant tuberculosis (TB) consists of a long, arduous, and complex treatment regimen for several months and in some cases years [1]. For over fifty years now, the mainstay for treating such patients with drug-resistant TB has been the administration of an array of antimicrobials [2]. Unfortunately, long-term use of these antimicrobials has been well-characterised and is associated with numerous toxic side-effects [1,2]. Due to the protracted course of treatment, medication compliance is low, resulting in increased incidences of drug resistant strains. Host-directed therapies (HDTs), used adjunctively with antimicrobials, may promote better bacterial clearance, thereby reducing the drug regimen duration which may increase compliance; limiting the development of drug-resistant TB and reducing toxic side-effects caused by long term exposure to antimicrobials.

Mycobacterial strains, including *Mycobacterium tuberculosis* (Mtb) and *Bacillus Calmette-Guérin* (BCG), thrive in iron-rich environments and require iron for optimal function and survival [3,4,5]. Targeting genes central to iron homeostasis in mycobacterial strains also attenuates their growth and function [3,6,7]. The role of iron metabolism in underpinning immunometabolic and effector responses in host immune cells provides wide scope for the development of new therapies against a variety of infectious diseases [8]. Iron chelators are administered for the treatment of conditions such as hereditary hemochromatosis and thalassemia [9] and can modulate Mtb growth and viability in human macrophages [10]. We previously hypothesised that the use of iron chelators could have multifaceted effects on immunometabolic function and could be utilised as a potential HDT to boost host immune responses during Mtb infection [11]. Indeed, acute iron deprivation with the iron chelator deferiprone reprograms primary human monocyte-derived macrophages (hMDMs) to increase glycolysis [12]. In a subsequent study, we examined whether the iron chelator, desferrioxamine (DFX), could support the function of hMDMs infected with Mtb [13]. Specifically, we demonstrated that DFX enhanced glycolytic metabolism and supported innate immune function in Mtb-infected hMDMs in a hypoxia-inducible factor 1-α dependent manner [13]. Accordingly, DFX’s ability to enhance glycolysis and innate immune function early in infected immune cells holds potential as an HDT, particularly in combination with current TB antimicrobials.

In the current study, we assessed if manipulating iron levels in macrophages infected with mycobacteria offered some insight into improving second-line antimicrobials that are widely used to treat drug-resistant tuberculosis. We investigated if DFX can support the antibacterial function of BCG-infected human macrophages treated with an array of second-line antimicrobials including moxifloxacin, bedaquiline, amikacin, clofazimine, linezolid and cycloserine. We provide some evidence to suggest that iron chelators may prove an effective adjunctive therapy in combination with current tuberculosis antimicrobials.

## 2. Results

### 2.1. Determining Sub-Optimal Concentrations of the Antimicrobials Moxifloxacin, Bedaquiline, Amikacin, Clofazimine, Linezolid and Cycloserine in hMDMs Infected with BCG

The bactericidal and bacteriostatic properties of the second-line drugs moxifloxacin, bedaquiline, amikacin, clofazimine, linezolid and cycloserine against a variety of TB strains has been widely documented in the literature [14,15,16,17,18]. Prior to examining the effect of DFX on the efficacy of these antimicrobials, it was first necessary to determine the optimal concentrations of these antimicrobials in primary human macrophages infected with BCG. Considering that these antimicrobials are known to exhibit very high efficacy against BCG, and we will seek to increase their efficacy further with DFX, it was necessary to identify sub-optimal concentrations of these antimicrobials to allow us to yield detectable levels of BCG inhibition. hMDMs were infected with BCG for three hours, unphagocytosed extracellular bacteria were washed away with PBS and hMDMs treated with varying concentrations of moxifloxacin, bedaquiline, amikacin, clofazimine, linezolid or cycloserine. Five days later, mycobacterial growth inhibition assays (MGIA) were carried out to investigate the ability of the drugs to control the growth of BCG (Figure 1). Antimicrobials deemed to increase the time to positivity (TTP) in these MGIA assays indicated a bactericidal or bacteriostatic effect; the abrogation of BCG growth is also plotted as a percentage change in TTP (or % change TTP). We found that 12.5 µg/mL moxifloxacin, 1 and 5 µg/mL bedaquiline, 5 µg/mL amikacin, 1 µg/mL clofazimine and 5 µg/mL linezolid significantly increased TTP in BCG-infected hMDMs (Figure 1A–E). Compared to controls, this corresponded to a significant reduction in BCG growth, as illustrated by percentage change in TTP, of 39.9% for 12.5 µg/mL moxifloxacin, of 50.4%, 57.9% and 94.7% for 0.5, 1 and 5 µg/mL bedaquiline, respectively, of 12.7% for 5 µg/mL amikacin, of 33.1% for 1 µg/mL clofazimine, and 9% for 5 µg/mL linezolid (Figure 1A–E). Cycloserine did not affect TTP in hMDMs infected with BCG (Figure 1F). The lowest antimicrobial concentration to significantly reduce TTP by >10% was chosen for further analyses in the rest of the current experiments. This cut-off threshold tallied with 12.5 µg/mL moxifloxacin, 0.5 µg/mL bedaquiline, 5 µg/mL amikacin and 1 µg/mL clofazimine.

Interestingly, 5 µg/mL linezolid only reduced TTP by 9%, and 10 µg/mL linezolid exhibited no significant effect on TTP or % change TTP (Figure 1E). Surprisingly, cycloserine did not have any significant effect on TTP or % change TTP. Linezolid and cycloserine are highly efficacious antimicrobials against the BCG TB strain in axenic conditions, however, we failed to recapitulate these observations in BCG-infected hMDMs. Accordingly, as some TB antimicrobials exhibit differential capacity to permeate lipid-rich cell membranes, we hypothesised that the reason why we did not observe increased bactericidal activity with linezolid and cycloserine in our model may be due to their inability to reach intracellular BCG. To test this, and the functionality of linezolid and cycloserine, hMDMs were infected with BCG for three hours, unphagocytosed BCG was washed off and the hMDMs were lysed to yield intracellular BCG. Intracellular BCG was then treated with 10 µg/mL linezolid or cycloserine in an axenic setting and MGIAs were carried out and TTP and % change TTP quantified as before. Treatment with linezolid (Appendix A) or cycloserine (Appendix A) increased TTP by 13.25 days and 2.6 days, consistent with a reduction in TTP of 348.9% and 58.9%, respectively, confirming the effectiveness of these two antimicrobials in an axenic setting. Consequently, as linezolid and cycloserine only reduced TTP in a cell free environment, these antimicrobials were excluded from further experiments.

### 2.2. Examining if the Iron Chelator, DFX, Can Increase the Efficacy of Moxifloxacin, Bedaquiline, Amikacin and Clofazimine in Primary hMDMs Infected with BCG

Next, we examined if the iron chelator, DFX, could help to support the bactericidal ability of the antimicrobials in hMDMs infected with BCG. To do this, hMDMs were infected with BCG for three hours and were subsequently treated with 100 µM DFX (as per our previous study [13]), in combination with the pre-determined concentrations of four second-line antimicrobials; 0.5 µg/mL bedaquiline, 5 µg/mL amikacin, 1 µg/mL clofazimine or 12.5 µg/mL moxifloxacin. Five days post infection, MGIAs were undertaken as before and TTP and % change TTP observed. Linezolid and cycloserine were not assessed due to their dampened ability to affect TTP in our BCG-hMDM model. We found that DFX treatment, in combination with bedaquiline, significantly increased TTP in hMDMs infected with BCG, coinciding with a significantly reduced % change TTP (Figure 2A). DFX did not affect TTP or % change TTP in hMDMs treated with amikacin (Figure 2B), clofazimine (Figure 2C) or moxifloxacin (Figure 2D). DFX treatment did not affect hMDM cell viability or relative hMDM cell numbers (Appendix A, respectively), as determined through propidium iodide-based cell exclusion and crystal violet assays, respectively.

### 2.3. Investigating the Effect of Combined DFX-Bedaquiline Treatment on Secreted Chemokine and Cytokine Levels in Primary hMDMs Infected with BCG

As dual DFX-bedaquiline treatment improved bacterial killing in hMDMs infected with BCG, we wanted to further examine this dual combination to gain some insight into how DFX could be supporting increased antibacterial activity in this model. To do this, we quantified the levels of various chemokines and cytokines, due to their ability to modulate the intracellular function and bactericidal abilities of various infected immune cells, including hMDMs. Some of these chemokines and cytokines have also been shown to recruit other immune cells to the site of infection in vivo thereby promoting mycobacterial clearance. To undertake this, we infected hMDMs with BCG for three hours and treated them as before with DFX in combination with bedaquiline. Twenty-four hours post BCG infection, cell supernatants were collected and the levels of the chemokines Eotaxin, Eotaxin-3, TARC, IP-10, MCP-1, MCP-4, MDC, MIP-1α and MIP-1β were quantified (Figure 3). No significant differences in the secreted levels of Eotaxin (Figure 3A), Eotaxin-3 (Figure 3B), TARC (Figure 3C), IP-10 (Figure 3D), MCP-1 (Figure 3E), MCP-4 (Figure 3F), MDC (Figure 3G), MIP-1α (Figure 3H) and MIP-1β (Figure 3I) were detected in DFX-bedaquiline treated hMDMs infected with BCG. Next, we quantified the levels of various cytokines in the same samples, including IFNγ, IL-6, TNF-α, IL-1β, IL-13, IL-2, IL-4, IL-10, IL12p70 and IL-8. Dual DFX-bedaquiline treatment significantly increased secreted levels of IFNγ (Figure 4A) IL-6 (Figure 4B) and IL-1β (Figure 4C) in hMDMs infected with BCG, compared to bedaquiline alone. However, no significant differences were detected for IL-8 (Figure 4D), IL-13 (Figure 4E), IL-2 (Figure 4F), IL-4 (Figure 4G), IL-10 (Figure 4H), IL12p70 (Figure 4I) or TNF-α (Figure 4J).

## 3. Discussion

The continued rise in resistance against TB antimicrobials has led to the need for HDT approaches to combat bacterial burden. Various FDA-approved drugs have been suggested in recent years as viable adjunctive HDTs for use in combination with current first-line and second-line TB antimicrobials. Among them, iron chelators have emerged as potential HDT candidates. Mtb requires iron for its survival and iron overload can exacerbate Mtb infection [19,20]. Iron chelators, such as DFX and silybin, can reduce Mtb viability in THP-1 monocytes [10]. Our group also recently showed that DFX enhances glycolytic metabolism in Mtb-infected hMDMs as well as supporting early innate immune function by enhancing transcript and protein levels of IL1β and TNF-α [13]. More recently, iron chelators such as DFX have undergone clinical trials for the treatment of COVID-19 [21]. While first-line TB antimicrobials are generally considered efficacious against drug-sensitive strains of TB, we hypothesised that the combined effects of DFX on host immune function along with the bactericidal effects of second-line TB antimicrobials could improve the treatment of drug-resistant TB infection. Our results indicate that DFX enhances the bactericidal potential of the second-line drug bedaquiline in primary hMDMs infected with BCG (Figure 4K), paving the way for future studies to examine the utility of iron chelators in combatting TB resistance.

Prior to assessing if DFX can increase the efficacy of second-line TB antimicrobials, sub-optimal concentrations of each antimicrobial needed to be determined first. This concentration was defined as the lowest concentration which significantly reduced time to positivity, or % change TTP, by at least 10%. We found that moxifloxacin, bedaquiline, amikacin and clofazimine significantly reduced % change TTP by at least 10% in BCG-infected hMDMs at the concentrations 12.5, 0.5, 5 and 1 µg/mL, respectively. Surprisingly, linezolid and cycloserine did not reduce bacterial burden intracellularly, as reflected by their inability to reduce % change TTP greater than 10%. We also confirmed in an axenic model that both linezolid and cycloserine are efficacious against the BCG strain, indicative that both drugs are impeded from penetrating hMDMs resulting in reduced bactericidal outcomes. In line with these findings, both linezolid and cycloserine have been shown to exert weak intracellular bactericidal activity against *S. aureus* and Mtb, respectively, in THP-1 macrophages [22,23]. This could help to explain the high doses of linezolid and cycloserine required clinically, often leading to adverse effects in patients with TB [23,24]. Indeed, high doses of linezolid results in various cytotoxic effects [2,25,26] while high concentrations of cycloserine can cause cognitive deterioration, dysarthria or psychotic crisis [2,27]. Thus, the need for alternative or adjunctive treatments to help reduce the dependence on these drugs is apparent.

Next, we examined if the cell-mediated effects of DFX on human macrophages could enhance the functionality of the second-line drugs bedaquiline, amikacin, clofazimine and moxifloxacin in hMDMs infected with BCG. Although ineffective in combination with amikacin, clofazimine and moxifloxacin, our findings demonstrate that DFX significantly reduces % change TTP compared to BCG-infected hMDMs treated with bedaquiline alone, thus supporting our hypothesis that DFX is an efficacious adjunct to TB antimicrobials. No study to our knowledge has previously shown enhanced bactericidal activity with an iron chelator and a TB antimicrobial in combination. Interestingly, bedaquiline-resistant TB cases are on an upward trajectory [28]. For example, it was recently discovered that a multi-drug resistant Mtb outbreak in Eswatini is associated with significantly higher resistance to bedaquiline [29]. More worryingly, this elevated bedaquiline resistance was undetectable through standard molecular drug susceptibility testing, specifically the Xpert MTB/RIF [29]. Thus, with bedaquiline-resistant TB cases on the rise, and with more specific and sensitive molecular drug susceptibility testing required to identify such TB cases, the need for improved HDTs for patients with multi-drug resistant TB is a necessity.

Finally, we investigated whether we could uncover any insight into how the DFX/bedaquiline combination may be increasing bactericidal activity in BCG-infected hMDMs. To aid this process, we examined the chemokine and cytokine profiles in matched supernatants from these hMDMs, as chemokine and cytokine secretion by hMDMs has been shown to modulate intracellular bactericidal activity [13,30,31,32,33]. Even though our data showed that DFX had no significant effect on the secretion of various chemokines and cytokines in BCG-infected hMDMs treated with bedaquiline, we found that the addition of DFX significantly increased secreted levels of IFN-γ, IL-6 and IL-1β. mRNA and protein levels of IL-1β have previously been shown to be increased by DFX in hMDMs infected with Mtb by our group [13]. Like IL-1β, it is generally considered that IFN-γ exerts an early protective immune response against Mtb infection through various cellular processes [31,32], such as activating antimicrobial activities like the production of reactive nitrogen intermediates [34]. More pertinently, it has also been shown that IFN-γ improves macrophage function in hMDMs from patients with MDR-TB [35], further leading to proposals for IFN-γ to be considered as a HDT for patients with TB. Although controversy exists around the ability of macrophages to produce IFN-γ, our lab has previously reported that murine and human macrophages can produce IFN-γ, albeit in small concentrations [36], as reported here in hMDMs. To further enhance the iron sequestration effect of DFX in the current model, IFN-γ could also directly contribute to reduced intraceullar iron availability and help to limit intracellular siderophilic bacterial replication, as evidenced by a recent study in human THP-1 cells infected with *L. monocytogenes*, *S. enterica* and *M. bovis* BCG [5]. This effect of IFN-γ is thought to be mediated by the upregulation of the iron exporter ferroportin and by the secretion of the hepcidin protein, which normally functions to inhibit iron export [5]. DFX has also been shown to increase IFN-γ receptor chain 1 and 2 expression in the cancer cells lines HCC, HuH7 and SNU449 [37]. Therefore, further investigations into the expression of IFN-γ and the IFN-γ receptors may help to highlight the mechanism of action of DFX and bedaquiline and if the combination can modulate intracellular bactericidal activity in primary hMDMs through IFN-γ signaling.

We also found that DFX significantly increased secreted levels of IL-6 in BCG-infected hMDMs treated with bedaquiline. IL-6 is an important, multi-functional cytokine, which regulates the immune response to TB infection [30]. IL-6 is involved in enhancing the early pro-inflammatory response to TB as well as promoting T-cell and B-cell induction in the later infection stages [38]. Interestingly, IL-6 has been shown to potentiate early IFN-γ production during Mtb infection [39]. Conversely, studies have shown that Mtb-induced IL-6 inhibits macrophage responses to IFN-γ [40] and can inhibit IFN-γ-induced autophagy in THP-1 cells [41]. Thus, deciphering the role IL-6 plays in the current model may offer additional insight into augmenting bactericidal activity in future studies. For example, if IL-6 imparts an inhibitory role in the current model, targeting IL-6 may increase bactericidal activity further. Moreover, due to the ability of bedaquiline to compartmentalise within subcellular regions of macrophages, DFX may also traffic to the same subcellular locations contributing to this synergistic effect [42,43]. As one of the mechanisms of resistance to bedaquiline are mutations leading to the overexpression of the MmpS5-MmpL5 efflux system, a pump naturally involved in siderophore transport, this could indicate that bedaquiline is not only subject to MmpS5-MmpL5 efflux, but it could also interfere with iron metabolism [44]. Furthermore, future studies should investigate the effect of other iron chelators in combination with TB antimicrobials. For example, iron-bound DFX is hydrophilic and a poor penetrator of lipid membranes [45]. Thus, improved efficacy may be observed in iron chelators with hydrophobic properties, such as silybin or deferiprone [10,12]. Studies should also examine the effect of these antimicrobials on host cellular metabolism, specifically glycolysis and oxidative phosphorylation, in the absence of live bacteria due to the ability of live TB-causing bacteria to directly subvert host metabolism [46,47]. Interestingly, bedaquiline has been shown to reprogram cell function in Mtb-infected human macrophages, by increasing macrophage bactericidal activity and modulating the expression of 1495 genes involved in metabolism, lysosome biogenesis and acidification [48]. Nonetheless, these data provide an in vitro proof-of-concept that iron chelators may prove an effective adjunctive therapy in combination with current TB antimicrobials.

## 4. Materials and Methods

### 4.1. hMDM Cell Culture

PBMCs were isolated from peripheral blood buffy coats (obtained from the Irish Blood Transfusion Services in Dublin, Ireland) by density gradient centrifugation with Lymphoprep^TM^ (Stemcell Technologies, Vancouver, British Columbia, Canada). PBMCs were seeded at 2.5 × 10^6^ cells/mL in Roswell Park Memorial Institute (RPMI) 1640 medium (Bio- Sciences Limited, Dublin, Ireland), supplemented with 10% AB-human serum (Sigma- Aldrich, St. Louis, Missouri, United States) and plated onto non-treated cell culture plates (Corning, Corning, New York, United States). LabTeks^TM^ (Nunc, Roskilde, Denmark) were also seeded to determine the multiplicity of infection (MOI) (see section “Infection of hMDMs with M. Bovis BCG”). To obtain hMDMs, the cells were cultured over 7 days at 37 °C and 5% CO_2_ to allow differentiation prior to experimentation. Cells were washed every 2–3 days to remove non-adherent cells.

### 4.2. Infection of hMDMs with M. Bovis BCG

M. bovis BCG was grown to log phase in Middlebrook 7H9 broth (Becton Dickinson, Franklin Lakes, New Jersey, United States) supplemented with albumin-dextrose-catalase (ADC; Becton Dickinson). On the day of infection, BCG was centrifuged at 3900 rpm for 10 min and resuspended in RPMI 1640 medium (supplemented with 10% AB-human serum). The suspension was passed 10 times through a 25-gauge needle and centrifuged at 800 rpm for 3 min in order to remove any bacterial clumps.

The volume of bacterial suspension required for a given MOI was determined by treating macrophages with a range of volumes of resuspended BCG. hMDMs in LabTeks^TM^ were incubated with BCG for 3 h, washed with pre-warmed PBS to remove extracellular bacteria and fixed with 2% paraformaldehyde (PFA; Sigma-Aldrich) for 10 min. hMDMs were then stained with Modified Auramine- O stain and Modified Auramine-O decolorizer (Scientific Device Laboratory, Des Plaines, Illinois, United States) followed by Hoechst 33242 (Sigma-Aldrich) to counterstain the nuclei. The cells were analysed under an inverted fluorescent microscope (Olympus IX51) to determine the average number of phagocytosed bacilli per cell and percentage of cells infected.

The required volume of bacilli was determined. Phagocytic variation between donors was accounted for by calculating the MOI for each donor (1–4 bacilli/cell, 40% positivity approximately). Cells were infected with the calculated volume of BCG. Three hours post-infection, unphagocytosed extracellular bacteria were removed by washing with PBS and fresh complete RPMI was added. hMDMs were then treated with the appropriate antibiotic (moxifloxacin, bedaquiline, amikacin, clofazimine, linezolid or cycloserine), in the presence or absence of DFX (100 μM). Macrophages were incubated at the indicated time periods at 37 °C and 5% CO_2_. Appropriate controls were also assayed, including uninfected hMDMs (untreated, treated with antibiotic alone, DFX alone or antibiotic with DFX) and infected hMDMs (untreated, treated with antibiotic alone or DFX alone).

### 4.3. Estimating Cell Viability and Cell Count Using Propidium Iodide (PI) Based Cell Exclusion Assays and Crystal Violet Assays

hMDMs were infected with BCG, as described above, and treated with DFX. Cell viability was determined using a PI based cell exclusion assay. Cells were stained with 5 µg/mL PI (Sigma-Aldrich), 20 µg/mL Hoechst 33342 (Sigma-Aldrich) and 50 µg/mL Hoechst 33258 (Sigma-Aldrich) for 30 min at room temperature. Total cell numbers were detected via Hoechst staining of nuclei (Blue channel: Ex 390 nm/Em 430 nm) and dying/dead cells were identified via positivity for PI staining (Orange channel: Ex 544 nm/EM 588 nm), using the Lionheart™ FX Automated Microscope Imaging Gen 5 System (Bio-Tek, Winooski, Vermont, United States). Five fields of view were acquired per treatment per well.

Relative cell numbers were determined using a Crystal Violet assay. hMDMs were fixed with 1% glutaraldehyde (in PBS) for 15 min at room temperature and washed with PBS. hMDMs were then incubated at room temperature for 20 min with 0.1% crystal violet solution (in dH2O). Cells were washed with dH2O and air dried overnight. Then, 1% Triton-X solution (in PBS) was added and the plate gently agitated for 15 min. The solution was then transferred to a 96-well plate, read at 590 nm on a spectrophotometer and plotted to determine relative cell numbers.

### 4.4. Mycobacterial Growth Inhibition Assay (MGIA)

PBMCs were isolated and differentiated into hMDMs in 12-well plates, infected with BCG and treated with antibiotics and DFX as described above. Twenty-four hours post BCG infection, 25% of the cell supernatant was removed and the cells supplemented with 25% fresh cRPMI. Five days post infection, a mycobacterial growth inhibition assay (MGIA) was employed to investigate the capacity to control the growth of BCG in response to the second-line TB antimicrobials. The cell monolayer was washed twice with PBS and replaced with 1% Triton X (in PBS). Macrophages were then harvested by cell scraping. The cell suspension in each well was passed 10 times through a 25-gauge needle and injected into labelled BACT/ALERT MP Culture Bottles (Biomerieux, Marcy-l’Étoile, France), supplemented with Lyophilized Antimicrobial supplement and Nutrient Supplement (Biomerieux). The culture bottles were then placed into the MGIA system (BACT/ALERT^®^ 3D; Biomerieux) and monitored until TTP (time to positivity) was reached. The TTP values were noted for each sample and % change TTP values calculated. % change TTP was determined for each sample using the following Equation (1):(1)$$\left(\frac{BCG\;TTP^{Day5}\;-\;Sample\;TTP^{Day5}}{BCG\;TTP^{Day5}}\right)\times 100$$
where ‘*BCG TTP^Day5^*’ is the TTP of BCG-infected hMDMs alone and ‘*Sample TTP^Day5^*’ is the TTP of the sample of interest (such as BCG-infected hMDMs treated with a TB drug alone or a TB drug and DFX). All moxifloxacin and clofazimine drug titration experiments were undertaken prior to Biomerieux introduced their lyophilized antimicrobial supplement and nutrient supplement system; Biomerieux supplied their BacT bottles without nutrient supplementation, which had to be purchased separately and added to BacT bottles on the day of the experiment. This change coincided with TTP values starting lower (from approximately 4). To account for this change, we normalised our TTP data to ‘% change TTP’, allowing it to be comparable across all drug/treatment groups (the TTP, as an absolute value, does not affect the data as all experiments are internally controlled and internally analysed).

### 4.5. Ex Vivo MSD Multiplex ELISA Analysis

PBMCs were cultivated and differentiated into hMDMs, stimulated with BCG and treated with bedaquiline and DFX as well as appropriate controls as described above. Supernatants collected at 24 h post infection were screened for the levels of cytokines/chemokines according to manufacturers’ instructions (Meso Scale Discovery Multi-Array technology, Rockville, Maryland, United States). The chemokines assessed included Eotaxin, Eotaxin-3, TARC, IP-10, MCP-1, MCP-4, MDC, MIP-1α and MIP-1β. The cytokines assessed included IFN-γ, IL-10, IL-12p70, IL-1β, IL-23, IL-2, IL-4, IL-6 and TNF-α.

### 4.6. Statistical Analysis

Data were analysed using Graph Pad Prism software version 9 (Graph Pad Prism, San Diego, CA, USA). Mixed effects REML and Friedman ANOVA tests with Dunnett’s/Dunn’s multiple comparisons tests were used to statistically analyse the optimisation of moxifloxacin, bedaquiline, amikacin, clofazimine, linezolid and cycloserine concentrations in hMDMs infected with BCG. Two-way repeated measures ANOVA tests with Šídák’s multiple comparisons were utilised to statistically analyse differences in %TTP (and % change TTP). Meso Scale Discovery Multi-Array technology chemokine and cytokine assays were statistically analysed using two-way repeated measures ANOVA tests with Šídák’s multiple comparisons tests. Differences of *p* < 0.05 (*), *p* < 0.01 (**) and *p* < 0.001 (***) were considered statistically significant.

## Figures and Tables

**Figure 1 ijms-22-02938-f001:**
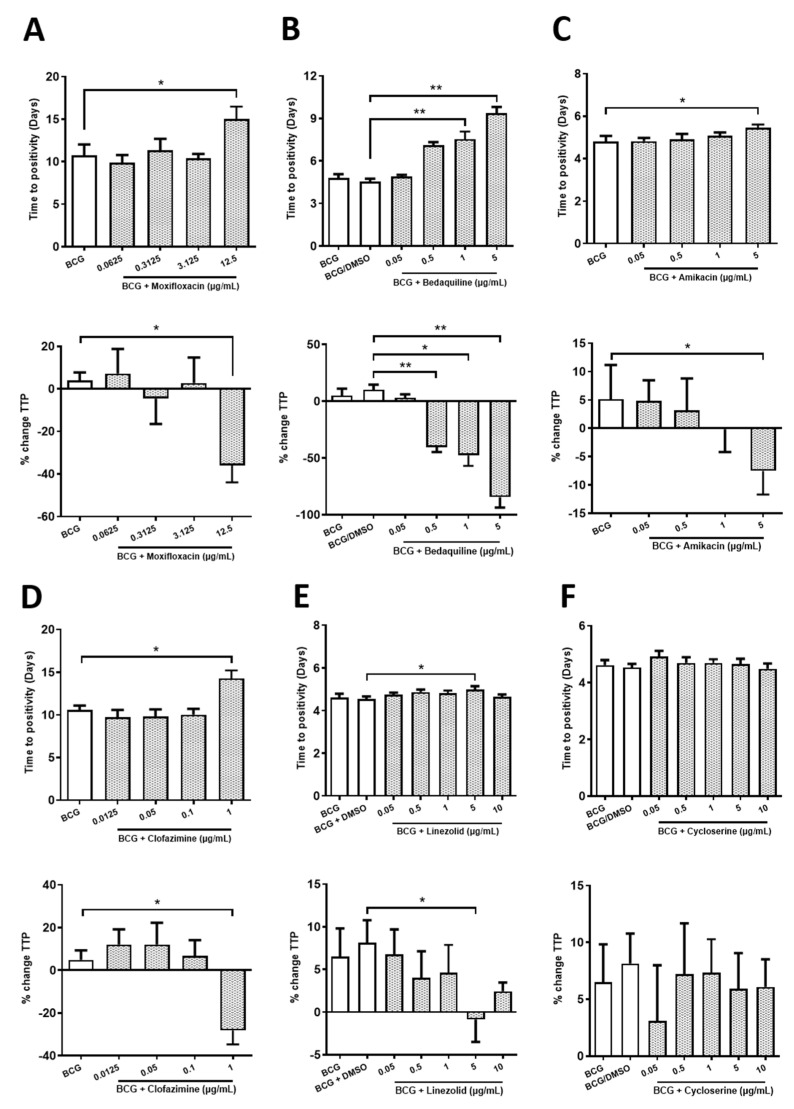
Determining the optimal concentration of moxifloxacin, bedaquiline, amikacin, clofazimine, linezolid and cycloserine in hMDMs infected with BCG. hMDMs, obtained from healthy blood donors, were infected with BCG for three hours and were subsequently treated with (**A**) moxifloxacin (0.0625, 0.3125, 3.125 and 12.5 µg/mL; *n* = 6), (**B**) bedaquiline (0.05, 0.5, 1 and 5 µg/mL; *n* = 3–4), (**C**) amikacin (0.05, 0.5, 1 or 5 µg/mL; *n* = 3–4), (**D**) clofazimine (0.0125, 0.05, 0.1 and 1 µg/mL; *n* = 4–5), (**E**) linezolid (0.05, 0.5, 1, 5 and 10 µg/mL; *n* = 3–7) or (**F**) cycloserine (0.05, 0.5, 1, 5 and 10 µg/mL; *n* = 3–7). Twenty-four hours post BCG infection, 25% of the cell supernatant was removed and the cells supplemented with 25% fresh cRPMI. Five days post infection, a mycobacterial growth inhibition assay (MGIA) was employed to investigate the capacity to control the growth of BCG in response to the second-line TB antimicrobials. Treatments deemed to increase time to positivity (TTP), shown in days, indicates a bactericidal or bacteriostatic effect; such inhibition in BCG growth can also be illustrated when plotted as percentage change in TTP (% change TTP; see methods section for description). The smallest antimicrobial concentration to significantly reduce TTP by >10% was chosen. Bars denote mean ± SEM. * *p* < 0.05 and ** *p* < 0.01 (mixed effects REML and Friedman ANOVA tests with Dunnett’s/Dunn’s multiple comparisons tests).

**Figure 2 ijms-22-02938-f002:**
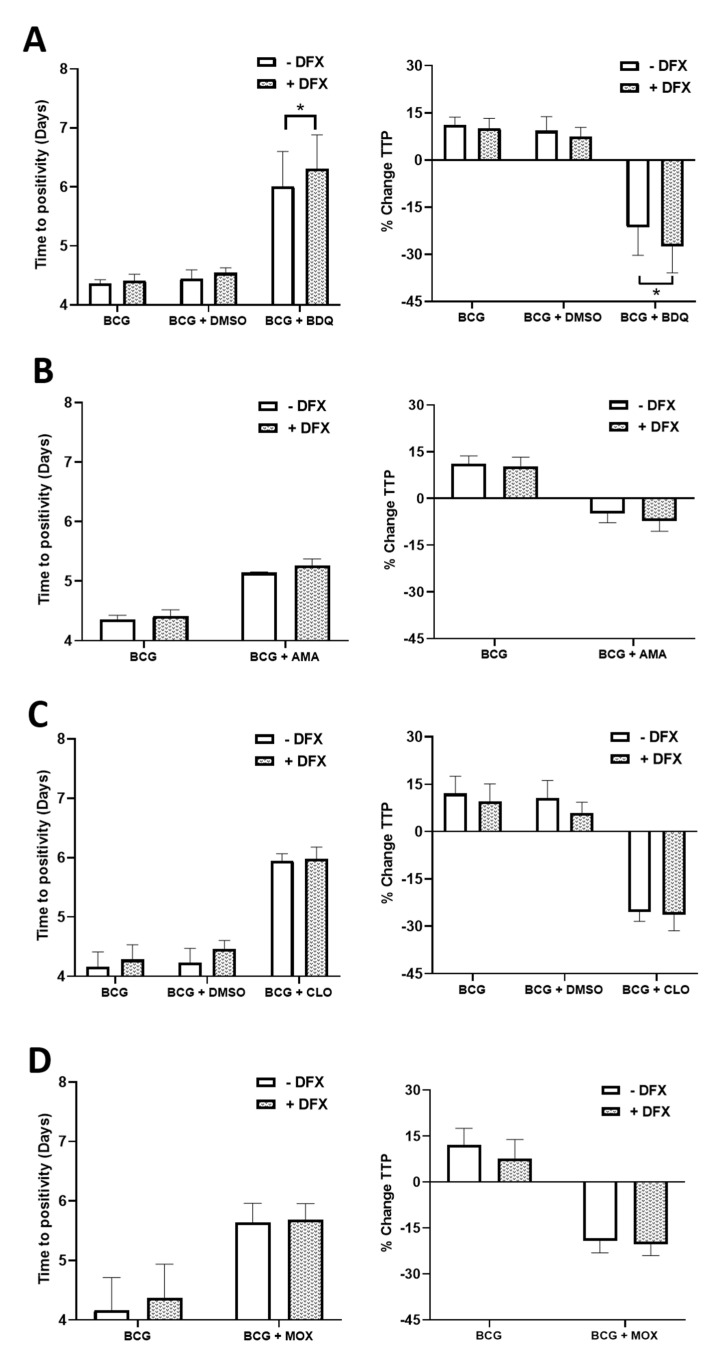
Examining if DFX can increase the efficacy of moxifloxacin, bedaquiline, amikacin and clofazimine in hMDMs infected with BCG. hMDMs, obtained from healthy blood donors, were infected with BCG for three hours and were subsequently treated with DFX (100 µM) in combination with (**A**) bedaquiline (0.5 µg/mL), (**B**) amikacin (5 µg/mL), (**C**) clofazimine (1 µg/mL) or (**D**) moxifloxacin (12.5 µg/mL). Twenty-four hours post BCG infection, 25% of the cell supernatant was removed, stored at −80 °C and the cells supplemented with 25% fresh cRPMI. Five days post infection, mycobacterial growth inhibition assays (MGIA) were undertaken to investigate the capacity to control the growth of BCG in response to the four antimicrobials. Treatments deemed to increase time to positivity (TTP), shown in days, indicates a bactericidal or bacteriostatic effect; such inhibition in BCG growth can also be illustrated when plotted as percentage change in TTP (% change TTP; see methods section for description). Bars denote mean± SEM. * *p* < 0.05 (Two-way repeated measures ANOVA tests with Šídák’s multiple comparisons tests, *n* = 5).

**Figure 3 ijms-22-02938-f003:**
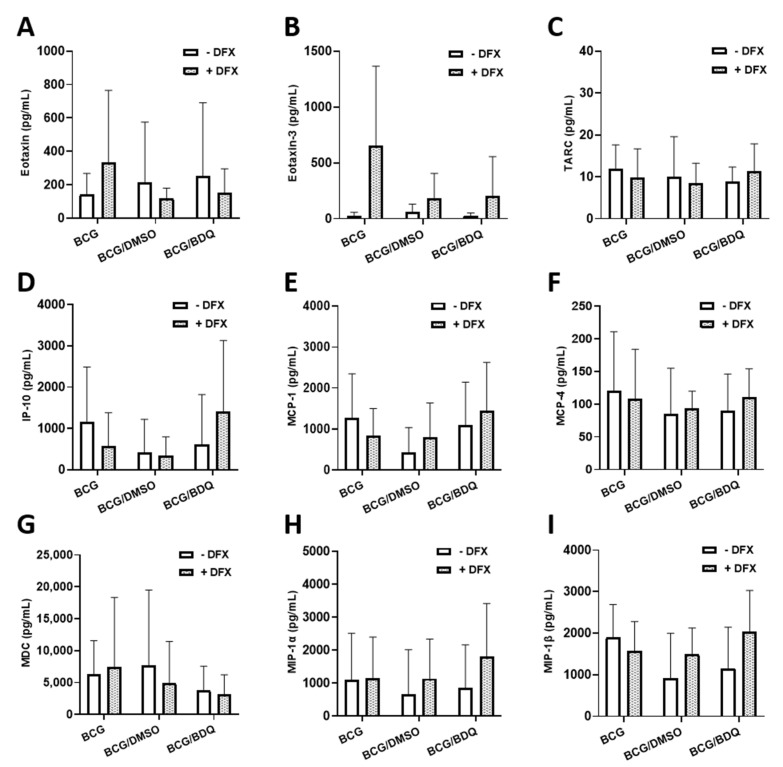
**Assessing if dual DFX-bedaquiline treatment affects chemokine secretions in BCG-infected hMDMs.** hMDMs, obtained from healthy blood donors, were infected with BCG for three hours and were subsequently treated with DFX (100 µM) in combination with bedaquiline (0.5 µg/mL). Twenty-four hours post BCG infection, cell supernatants were collected and levels of (**A**) eotaxin, (**B**) eotaxin-3, (**C**) TARC, (**D**) IP-10, (**E**) MCP-1, (**F**) MCP-4, (**G**) MDC, (**H**) MIP-1α and (**I**) MIP-1β were quantified using Meso Scale Discovery Multi-Array technology (*n* = 5). Bars denote mean ± SEM. *p* > 0.05 (Two-way repeated measures ANOVA tests with Šídák’s multiple comparisons tests).

**Figure 4 ijms-22-02938-f004:**
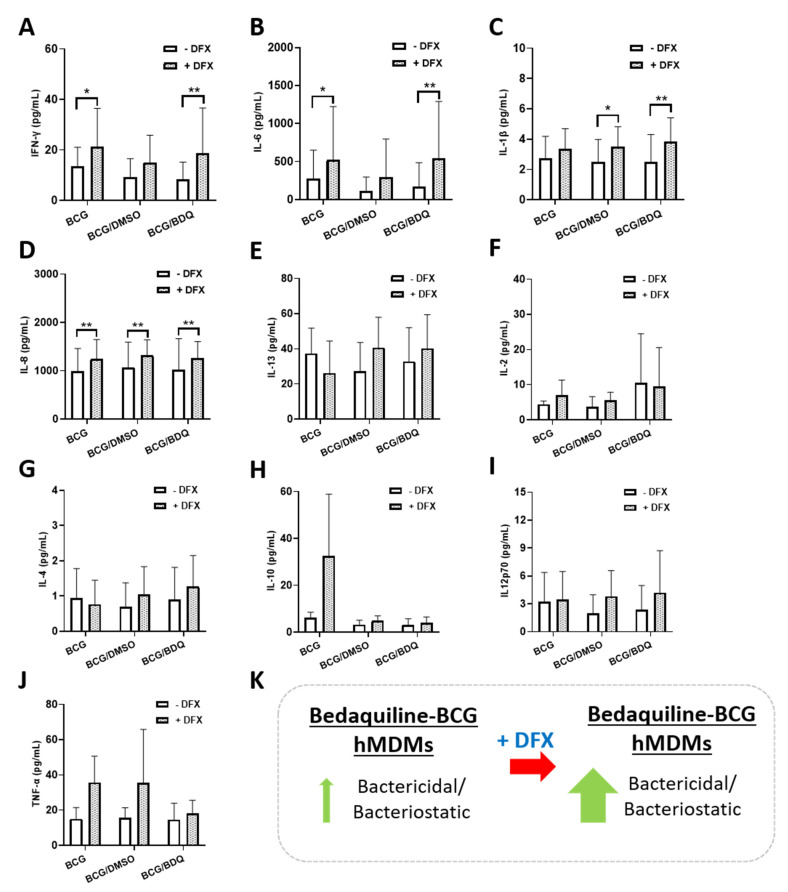
**Determining the effect of combined DFX-bedaquiline treatment on cytokine secretions in hMDMs infected with BCG.** hMDMs, obtained from healthy blood donors, were infected with BCG for three hours and were subsequently treated with DFX (100 µM) in combination with bedaquiline (0.5 µg/mL). Twenty-four hours post BCG infection, cell supernatants were collected and levels of (**A**) IFNγ, (**B**) IL-6, (**C**) IL-1β, (**D**) IL-8, (**E**) IL-13, (**F**) IL-2, (**G**) IL-4, (**H**) IL-10, (**I**) IL12p70 and (**J**) TNF-α assayed using Meso Scale Discovery Multi-Array technology (*n* = 5). (**K**) DFX increases the bactericidal/bacteriostatic properties of bedaquiline in primary hMDMs infected with BCG. Bars denote mean ± SEM. * *p* < 0.05 and ** *p* < 0.01 (Two-way repeated measures ANOVA tests with Šídák’s multiple comparisons tests).

## Data Availability

Data is contained within the article or Appendix A.

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
