# Peer review of "The Iron Chelator Desferrioxamine Increases the Efficacy of Bedaquiline in Primary Human Macrophages Infected with BCG"

_ijms, 2021, doi:10.3390/ijms22062938_

Round 1
Reviewer 1 Report
This is the in vitro study regarding if using iron chelator, desferrioxamine, can support the function of human macrophages infected with BCG, treated with an array of second-line antimicrobials, including moxifloxacin, bedaquiline, amikacin, clofazimine, linezolid, and cycloserine, and concluded that iron chelators may prove an effective adjunctive therapy for bedaquiline in combination with current tuberculosis antimicrobials. The concept is innovative and the study design is well. The study provides a new concept about treatment for tuberculosis but it needs more human study for practical use in the medical field.
Reviewer 2 Report
The manuscript by Cahill et al. describes the possible use of iron chelator desferrioxamine (DFX) as a host-directed therapeutic to increase the efficiency of anti-tuberculosis drugs. The authors state that they observed that DFX increases the efficiency of bedaquiline in ex vivo macrophages model infected with BCG. This may be a very interesting finding, however the results section of the manuscript needs a major revision in order to be clear and informative, prior to acceptance.
Major comments:
Schemes 1 and 2 are not mentioned within the text. The difference between schemes and figures is not clear. Maybe they can be named as figures and mentioned in the text?
Section 2.1 is overloaded with numbers, some of which are better suited for materials and methods (antibiotics concentrations), while others are represented by the figure 1.
The major concern about figure 1 is the difference in TTP in the control sample: the average TTP for BCG varies between experiments from 4 to 10 days. This is a big spread in growth time. Is this method reproducible? This should be commented in the text. Otherwise measuring a 10% change in TTP seems unacceptable.
Figure 2A: it is not clear from the histogram, how BCG+BDQ+DFX has a significant difference from BCG+BDQ.
Figures 3 and 4 are also overloaded with information. Moreover it is not clear how in some cases authors observe significant changes and in others do not. DFX itself apparently alters the level of cytokines and interleukins in hMDMs, as a HDT should probably do. Either the figures are not informative, or their interpretation is incorrect.
Minor comments:
The end of introduction section (from “We found that adjunctive DFX-bedaquline…” to the end) may be a better fit for “conclusion” section.
Please add line numbers in the manuscript.
One of the mechanisms of resistance to bedaquiline are mutations leading to overexpression of the MmpS5-MmpL5 efflux system [10.1128/AAC.00037-14]. This pump is naturally involved in siderophore transport, but has shown to be able to provide resistance to multiple drugs in mycobacteria. Could this mean that bedaquiline may not only be subjected to MmpS5-MmpL5 efflux, but also interfere with iron metabolism? I think this may be worth discussing in the discussion section of the manuscript.
Round 2
Reviewer 2 Report
The authors have managed to significantly improve the manuscript during the revision. The points that were not clearly presented in the first version are now easily understood. I would recommend this manuscript for acceptance after a minor revision.
Minor comments:
Though the authors have now added lines 422-424 about the Biomerieux antimicrobial supplement, I would recommend adding additional info in the manuscript about how this affected the experiment (increased TTP), as it was done in the response letter.
The authors also state in the response letter that the “new analyses have also uncovered significantly higher levels of IL-8 and IL-1β in BCG-infected hMDMs co-treated with BDQ/DFX (figure 4C-4D)”. However the values of IL-8 (figure 4D in manuscript v2) are in the range of 1000-1200 pg/mL, while in manuscript v1 (figure 4J) they were 10000-30000 pg/mL (10-15 times higher). Please double-check that the revised values are correct.
